# The Design and Immunogenicity of an HIV-1 Clade C Pediatric Envelope Glycoprotein Stabilized by Multiple Platforms

**DOI:** 10.3390/vaccines13020110

**Published:** 2025-01-22

**Authors:** Sanjeev Kumar, Iván del Moral-Sánchez, Swarandeep Singh, Maddy L. Newby, Joel D. Allen, Tom P. L. Bijl, Yog Vaghani, Liang Jing, Rakesh Lodha, Eric A. Ortlund, Max Crispin, Anamika Patel, Rogier W. Sanders, Kalpana Luthra

**Affiliations:** 1Department of Medical Microbiology and Infection Prevention, Amsterdam UMC, University of Amsterdam, 1105 AZ Amsterdam, The Netherlands; sanjeev.kumar@emory.edu (S.K.); i.delmoralsanchez@amsterdamumc.nl (I.d.M.-S.); t.p.bijl@amsterdamumc.nl (T.P.L.B.); 2Department of Biochemistry, All India Institute of Medical Sciences, New Delhi 110029, India; swarandeepbiologist@gmail.com; 3Department of Pediatrics, Division of Infectious Diseases, Emory Vaccine Center, Emory University School of Medicine, Atlanta, GA 30322, USA; 4Amsterdam Institute for Infection and Immunity, 1105 AZ Amsterdam, The Netherlands; 5School of Biological Sciences, University of Southampton, Southampton SO17 1BJ, UK; m.l.newby@soton.ac.uk (M.L.N.); joel.allen@soton.ac.uk (J.D.A.); max.crispin@soton.ac.uk (M.C.); 6Department of Biochemistry, Emory University School of Medicine, Atlanta, GA 30322, USA; yog.suresh.vaghani@emory.edu (Y.V.); liang.jing@emory.edu (L.J.); eortlun@emory.edu (E.A.O.); anamika.patel@emory.edu (A.P.); 7Department of Pediatrics, All India Institute of Medical Sciences, New Delhi 110029, India; rlodha1661@gmail.com; 8Department of Microbiology and Immunology, Weill Medical College of Cornell University, New York, NY 10065, USA

**Keywords:** HIV-1, neutralizing antibodies, rabbit immunization

## Abstract

Background: Elite-neutralizer-derived HIV-1 envelopes (Envs), which induce broadly neutralizing antibodies (bnAbs), can inform HIV-1 vaccine design by serving as templates for bnAb-eliciting vaccines. Since single Env-based immunizations are insufficient to induce bnAb responses, sequential regimens using multivalent immunogens or Env cocktails hold greater promise. This underscores the need to develop stable Env trimers from diverse HIV-1 strains, particularly clade-C, which accounts for 50% of global infections and over 90% in India and South Africa. While various platforms exist to stabilize soluble Env trimers for use as antigenic baits and vaccines, stabilizing clade C trimers remains challenging. Methods: We stabilized an HIV-1 clade C trimer based on an Env isolated from a pediatric elite neutralizer (AIIMS_329) using multiple platforms, including SOSIP.v8.2, ferritin nanoparticles (NPs) and I53-50 two-component NPs, followed by characterization of their biophysical, antigenic, and immunogenic properties. Results: The stabilized 329 Envs showed binding to multiple HIV-1 bnAbs, with negligible binding to non-neutralizing antibodies. Negative-stain electron microscopy confirmed the native-like conformation of the Envs. Multimerization of 329 SOSIP.v8.2 on ferritin and two-component I53-50 NPs improved the affinity to HIV-1 bnAbs and showed higher immunogenicity in rabbits. Conclusions: The soluble 329 Env protein could serve as an antigenic bait, and multimeric 329 NP Envs are potential vaccine candidates.

## 1. Introduction

Clade C accounts for about 50% of HIV-1 infections worldwide and over 90% of infections in India and South Africa [1,2]. HIV-1 envelopes (Envs) from elite neutralizers that develop broadly neutralizing antibodies (bnAbs) can guide HIV-1 vaccine design by serving as templates to induce bnAb responses through vaccination [3,4,5,6,7]. Stabilized native-like HIV-1 Env soluble trimers, mainly from non-clade C isolates, have shown the ability to induce neutralizing antibody (nAb) responses in animal models [8,9,10,11,12]. These native-like Envs have also been used as antigenic baits to isolate exceptionally potent second-generation HIV-1 bnAbs from both adults and children [13,14,15,16,17]. However, the instability and low expression levels of soluble HIV-1 clade C Env trimers have made it challenging to develop clade C Env-based vaccines that induce protective bnAb responses [18,19,20]. Since single Env-based regimens are unlikely to generate bnAb responses, sequential immunizations using multivalent immunogens or Env cocktails could offer more promise [6,21]. This highlights the need to generate stable Env trimers from diverse HIV-1 strains and clades, particularly those from elite neutralizers that develop exceptionally potent bnAbs with multi-epitope specificities [4,20,22,23].

SOSIP mutations (SOS refers to a disulfide bond between the gp120 (501C) and gp41 (605C) region, and IP refers to the I559P mutation) are widely known to produce native-like soluble HIV-1 Env trimers [8,9,24,25]. However, the use of these mutations for the design of clade C Envs has mostly yielded low expressing Envs with poor antigenicity and immunogenicity [19,20]. Recent studies have described mutations that improve the yield, antigenicity, and stability of soluble Env proteins, including some clade C Envs [10,11,26,27,28,29,30,31]. Most of these mutations were developed using the clade A BG505 Env [9,29,30,31], but their effectiveness on other clades remains limited. Moreover, a limited number of HIV-1 clade C native-like Envs from India have been studied [19,20], but none from pediatric elite neutralizers [13,22,23,32].

Previously, we reported HIV-1 Env sequences from a pair of Indian clade C chronically infected pediatric elite neutralizer monozygotic twins (AIIMS_329 and AIIMS_330) whose plasma exhibited exceptionally strong bnAb responses with multiple-epitope specificities against a large panel of multi-clade heterologous Env pseudoviruses [13,22]. Such Envs have the potential to serve as templates for immunogen design and as useful antigenic baits to isolate HIV-1 bnAbs for immunotherapeutic use. Notably, the BG505 HIV-1 Env sequence, isolated from an infant transmitted founder virus, has been extensively studied, with several BG505-derived native-like Env trimers currently being evaluated in vaccine clinical trials [6,12,21,24,28]. In the past few years, using BG505 trimers as antigenic bait, several second-generation potent HIV-1 bnAbs have been identified in both adults and children [13,14,15,16,33].

Herein, we designed and characterized a soluble HIV-1 clade C_329 Env trimer, derived from a circulating virus in an Indian pediatric elite neutralizer AIIMS_329, by stabilizing the sequence using SOSIP v8.2 [24], displaying it on ferritin [34], and self-assembling two-component I53-50 nanoparticles (NPs) [27,35]. The stabilized native-like 329 SOSIP.v8.2 Env trimer showed strong binding to key HIV-1 bnAbs and negligible binding to non-neutralizing antibodies (non-nAbs). Multimerization of 329 SOSIP.v8.2 trimers on ferritin and two-component I53-50 NPs improved the affinity to HIV-1 bnAbs and demonstrated a higher immunogenicity in rabbits compared to 329 SOSIP.v8.2 Env trimers. The native-like conformation of the proteins was confirmed using low-resolution negative-stain electron microscopy (nsEM) and cryo-electron microscopy (cryoEM). We successfully stabilized and demonstrated the immunogenicity of varied versions of an Indian clade C native-like Env trimer derived from a pediatric elite neutralizer. These trimers could be valuable antigenic baits for HIV-1 bnAb discovery and could serve as components of multivalent HIV-1 vaccines and as tools for evaluating vaccine responses in the future. Multimeric antigen presentation has evolved as a promising strategy to improve the overall stability and antigenicity of HIV-1 clade C and non-clade C trimeric Env glycoproteins.

## 2. Materials and Methods

### 2.1. Construct Design

The 329 *env* gene was derived from a previously identified Indian clade C HIV-1 sequence obtained from a pediatric elite neutralizer AIIMS_329 (329.14.B1, GenBank: MK076593.1), as previously described [22].

The 329 SOSIP.v8.2 construct was designed by introducing the SOSIP mutations (501C-605C, 559P), including a multibasic furin cleavage site (hexa-arginine or R6) between gp120 and gp41. This protein also incorporated TD8 (47D, 49E, 65K, 165L, 429R, 432Q, 500R) and MD39 stabilizing mutations (106E, 271I, 288L, 304V, 319Y, 363Q, 519S, 568D, 570H, 585H), as well as a mutation to reduce V3 exposure (66R)^21^. We also introduced changes to optimize the epitope of PGT145 (166R, 168K, 170Q, 171K) (Appendix A).

The 329.SOSIP.v8.2–ferritin construct was generated by fusing the N-terminus from *Helicobacter pylori* ferritin (Genbank accession no. NP_223316), starting from Asp5 to Ser167, to the SOSIP.664 C-terminus (truncated at position 664), separated by a Gly-Ser (GSG) linker, as described previously [28].

To create the 329-I53-50A.1NT1 construct, the original I53-50A.1NT1 plasmid was used, as described previously [27]. The modifications constitute the introduction of GSLEHHHHHH after the final residue to introduce a C-terminal histidine-tag.

All constructs comprised the above-described sequences preceded by a tissue plasminogen activator (tPA) signal peptide (MDAMKRGLCCVLLLCGAVFVSPSQEIHARFRRGAR). Untagged Env constructs presented a STOP codon after position 664. Strep-tagged SOSIP.v8.2 constructs included an additional Twin-Strep-Tag amino acid sequence (GSGGSSAWSHPQFEKGGGSGGGSGGSAWSHPQFEKG) after position 664.

All genes were codon-optimized for mammalian expression synthesized by Genscript (Piscataway, NJ, USA) and cloned by restriction–ligation into a pPI4 plasmid.

### 2.2. HIV-1 Envelope Protein Expression

SOSIP Env and SOSIP Env-NP fusion proteins were expressed as described previously [30]. Briefly, HIV-1 Env and furin protease-encoding plasmids were mixed in a 3:1 Env (450 µg) to furin (150 µg) ratio (*w*/*w*) and incubated with PEImax (Polysciences Europe GmBH, Eppelheim, Germany) in a 3:1 (*w*/*w*) PEImax to DNA ratio for 20 min at room temperature (RT). Subsequently, the transfection mixtures were added to the supernatant of HEK293F suspension cells (Invitrogen^TM^, Bleiswijk, The Netherlands, cat no. R79009) and maintained in FreeStyle Expression Medium (Gibco) at a density of 0.8–1.2 million cells/mL. Seven days post-transfection, the supernatants were harvested, centrifuged, and filtered using Steritops (0.22 µm pore size; Millipore, Amsterdam, The Netherlands) before protein purification.

### 2.3. HIV-1 Envelope Protein Purification

SOSIP Env and SOSIP Env-NP fusion proteins were purified by PGT145 bnAb immunoaffinity chromatography, as described previously [29]. Briefly, HIV-1 pre-fusion closed trimer-specific bnAb PGT145 [14,36] was covalently coupled to the CNBr-activated Sepharose 4B beads (GE Healthcare/Cytiva, Medemblik, The Netherlands) according to the manufacturer’s protocol for antibody coupling. The PGT145-coupled CNBr Sepharose 4B bead column was used to purify HIV-1 Env trimers or Env trimer NPs. Proteins contained in HEK293F filtered supernatants were captured on PGT145-coupled CNBr-activated sepharose 4B beads by means of overnight rolling incubation at 4 °C. Subsequently, the mixture of supernatants and beads was passed over Econo-Column chromatography columns (Biorad, Hercules, CA, USA). The columns were then washed with three column volumes of 0.5 M NaCl and 20 mM Tris HCl pH 8.0 solution. After elution with 3 M MgCl_2_ pH 7.5, the proteins were buffer-exchanged into TN75 (75 mM NaCl, 20 mM Tris HCl pH 8.0) or PBS buffer by ultrafiltration with Vivaspin20 filters (Sartorius, Göttingen, Germany) of MWCO 100 kDa. Protein concentrations were determined from the A280 values measured using a NanoDrop2000 device (Thermo Fisher Scientific, Bleiswijk, The Netherlands), and the molecular weight and extinction coefficient values were calculated by the ProtParam Expasy webtool.

### 2.4. I53-50B.4PT1 Protein Expression and Purification

I53-50B.4PT1 protein purification was performed as described previously [27]. Briefly, Lemo21 cells (DE3) (NEB), which were grown in LB (10 g Tryptone, 5 g Yeast Extract, 10 g NaCl) in 2 L baffled shake flasks or a 10 L BioFlo 320 Fermenter (Eppendorf, Framingham, MA, USA), were transformed with an I53-50B.4PT1-encoding plasmid. After inducing protein expression via the addition of 1 mM IPTG, the cells were subjected to shaking for 16 h at 18 °C. Microfluidization was used to harvest and lyse the cells, using a Microfluidics M110P machine at 18,000 psi in 50 mM Tris, 500 mM NaCl, 30 mM imidazole, 1 mM PMSF, and 0.75% CHAPS. Proteins were purified by applying clarified lysates to a 2.6 × 10 cm Ni Sepharose 6 FF column (Cytiva) on an AKTA Avant150 FPLC system (Cytiva). A linear gradient of 30 mM to 500 mM imidazole in 50 mM Tris, pH 8.0, 500 mM NaCl, and 0.75% CHAPS was used to elute both proteins. Next, the pooled fractions were subjected to size-exclusion chromatography on a Superdex 200 Increase 10/300 or a HiLoad S200 pg GL size-exclusion chromatography (SEC) column (Cytiva) in 50 mM Tris pH 8.0, 500 mM NaCl, and 0.75% CHAPS buffer. I53-50B.4PT1 was eluted at ~0.45 CV. Before nanoparticle assembly, protein preparations were tested to confirm low levels of endotoxin. To remove endotoxin, purified I53-50B.4PT1 was immobilized on Ni^2+^-NTA resin in a 5 mL HisTrap HP column (GE Healthcare) equilibrated with the following buffer: 25 mM Tris pH 8.0, 500 mM NaCl, and 0.75% CHAPS. Immobilized I53-50B.4PT1 was then washed with ~10 CVs of the equilibration buffer. The protein was eluted over a gradient to 500 mM imidazole in an equilibration buffer. Fractions containing I53-50B.4PT1, which eluted around ~175 mM imidazole, were concentrated in a Vivaspin filter with a 10 kDa molecular weight cutoff and subsequently dialyzed twice against the equilibration buffer (GE Healthcare).

### 2.5. HIV-1 SOSIP-I53-50NP Assembly

HIV-1 SOSIP-I53-50NP assembly was performed as described previously [27]. Briefly, after PGT145 bnAb column purification (see HIV-1 Env protein expression and purification), the SOSIP–component A fusion protein (329 SOSIP.v8.2-I53-50A.1NT1) was passed through a Superose 6 Increase 10/300 GL (GE Healthcare) SEC column in Assembly Buffer II (25 mM Tris, 500 mM NaCl, 5% glycerol pH 8.2) to remove aggregated proteins. The glycerol component was included in the Assembly Buffer II to minimize aggregation of the SOSIP–component A fusion proteins during the assembly of NPs, but we found that their presence increased the recovery of the assembled NPs during the concentration and dialysis stages described below. After the SEC procedure, the column fractions containing non-aggregated SOSIP-I53-50A.1NT1 proteins were immediately pooled and mixed in an equimolar ratio with I53-50B.4PT1 (produced as described above) for an overnight (~16 h) incubation at 4 °C. The assembly mix was then concentrated at 350× *g* using Vivaspin filters with a 10 kDa molecular weight cutoff and passed through a Superose 6 Increase 10/300 GL column in Assembly Buffer II (GE Healthcare). The fractions corresponding to the assembled NPs (elution between 8.5 and 10.5 mL with a peak at 9 mL) were pooled and concentrated at 350× *g* using Vivaspin filters with a 10 kDa molecular weight cutoff (GE Healthcare). The assembled NPs were then buffer-exchanged into phosphate-buffered saline (PBS) by dialysis at 4 °C overnight, followed by a second dialysis step for a minimum of 4 h, using a Slide-A-Lyzer MINI dialysis device (20 kDa molecular weight cutoff; Thermo Fisher Scientific). Nanoparticle concentrations were determined by using the Nanodrop method with the particles’ peptidic molecular weight and extinction coefficient. To obtain these values, first, the molecular weight and extinction coefficient of the SOSIP-I53-50A.1NT1 and I53-50B.4PT1 components were obtained by filling in their amino acid sequence in the online Expasy web server (ProtParam tool). The peptidic mass and extinction coefficient of SOSIP-I53-50NP was then calculated by summing the obtained peptidic masses and extinction coefficients, respectively, of each component of the NP.

### 2.6. SDS-PAGE and BN-PAGE Analyses

For SDS-PAGE and blue native polyacrylamide gel electrophoresis (BN-PAGE) analyses, 2 µg of SOSIP trimers, or equimolar amounts of SOSIP–ferritin (2.5 µg) and SOSIP-I53-50 (3.2 µg) NPs, was ran over Novex Wedge well 4–12% Tris-Glycine and NuPAGE 4–12% Bis-Tris and polyacrylamide gels (both from Invitrogen), respectively, as described previously [30]. Subsequently, the gels were run as per the manufacturer’s protocol and then stained with PageBlue Protein Staining Solution (Thermo Fisher Scientific, Bleiswijk, The Netherlands) or the Colloidal Blue Staining Kit (Invitrogen™, Bleiswijk, The Netherlands).

### 2.7. Enzyme-Linked Immunosorbent Assay (ELISA)

StrepTactinXT ELISAs were performed as described previously [29] with few modifications. StrepTactinXT coated microplates (IBA GmbH, Göttingen, Germany) do not require any functionalization or blocking steps before protein immobilization. Briefly, 100 µL of PGT145 bnAb column-purified 329 SOSIP protein in TBS (1 µg/mL) was dispensed in the corresponding wells for protein immobilization via a 2 h incubation at room temperature. The subsequent steps to measure the binding of the test antibodies were performed similarly to those previously described [30]. Briefly, following a double-wash step with TBS to remove unbound proteins, serial dilutions of the test primary antibodies (HIV-1 mAbs) or immunized rabbit sera in casein blocker were added and incubated for 2 h. After 3 washes with TBS, HRP-labeled goat anti-human IgG (Jackson Immunoresearch, West Grove, PA, USA) diluted at 1:3000 in casein blocker was added and incubated for 1 h, followed by 5 washes with TBS/0.05% Tween20. Plates were developed with the o-phenylenediamine substrate (Sigma-Aldrich, Amsterdam, The Netherlands, #P8787) in 0.05 M phosphate–citrate buffer (Sigma-Aldrich, #P4809) pH 5.0, containing 0.012% hydrogen peroxide (Thermo Fisher Scientific, Bleiswijk, The Netherlands, #18755). Absorbance was measured at 490 nm to obtain the binding curves.

### 2.8. Biolayer Interferometry (BLI)

The BLI assay was performed as described previously [30], using an Octet K2 (ForteBio, Bemmel, The Netherlands) device at 30 °C and 1000 rpm agitation. Briefly, test antibodies diluted in kinetics buffer (PBS/0.1% bovine serum albumin/0.02% Tween20) were loaded on protein A sensors (ForteBio, Bemmel, The Netherlands) at an interference pattern shift of 1 nm. Sensors were equilibrated in the kinetic buffer for 60 s to obtain a baseline before protein association. Subsequently, all the purified SOSIP trimers or trimeric NPs diluted in kinetics buffer (100 nM) were allowed to associate for 300 s and dissociate for 300 s. Binding data were pre-processed and exported using Octet software, version 9.0.

### 2.9. Nano Differential Scanning Fluorimetry (NanoDSF)

Protein thermostability was evaluated with a Prometheus NT.48 instrument (NanoTemper Technologies GmbH, Munich, Germany). Proteins at a concentration of 1 mg/mL were loaded to the grade capillaries, and the intrinsic fluorescence signal was measured while the temperature was increased by 1 °C/min, with an excitation power of 40%. The temperature of onset (T_onset_) and the temperature of melting (T_m_) were determined using Prometheus NT software NT.48.

### 2.10. Site-Specific Glycan Analysis Using Mass Spectrometry

First, 100 µg aliquots of each sample were denatured for 1 h in 50 mM Tris/HCl, pH 8.0, containing 6 M of urea and 5 mM dithiothreitol (DTT). Next, Env samples were reduced and alkylated by adding 20 mM iodoacetamide (IAA) and incubated for 1 h in the dark, followed by a 1 h incubation with 20 mM DTT to eliminate residual IAA. The alkylated Env samples were buffer-exchanged into 50 mM Tris/HCl, pH 8.0, using Vivaspin columns (10 kDa) and three of the aliquots were digested separately overnight using trypsin, chymotrypsin (Mass Spectrometry Grade, Promega, Leiden, The Netherlands), or alpha lytic protease (Sigma-Aldrich) at ratio of 1:30 (*w*/*w*), i.e., 1 µg enzymes for 30 µg Env protein. The next day, the peptides were dried and extracted using an Oasis HLB µElution Plate (Waters, Milford, MA, USA).

The peptides were dried again, re-suspended in 0.1% formic acid, and analyzed by nanoLC-ESI MS with an Easy-nLC 1200 (Thermo Fisher Scientific) system coupled to a Fusion mass spectrometer (Thermo Fisher Scientific) using stepped higher energy collision-induced dissociation (HCD) fragmentation. The peptides were separated using an EasySpray PepMap RSLC C18 column (75 µm × 75 cm). A trapping column (PepMap 100 C18 3 μM 75 μM × 2 cm) was used in line with the LC before separation with the analytical column. The LC conditions were as follows: 275 min linear gradient consisting of 0–32% acetonitrile in 0.1% formic acid over 240 min followed by 35 min of 80% acetonitrile in 0.1% formic acid. The flow rate was set to 300 nL/min. The spray voltage was set to 2.5 kV and the temperature of the heated capillary was set to 55 °C. The ion transfer tube temperature was set to 275 °C. The scan range was 375–1500 m/z. The stepped HCD collision energies were set to 15, 25, and 45%, and the MS2 for each energy was combined. Precursor and fragment detection were performed using an Orbitrap at a resolution MS1 = 100,000 and MS2 = 30,000. The AGC target for MS1 = 4 × 5^10^ and for MS2 = 5 × 4^10^, and the injection time for MS1 = 50 ms and for MS2 = 54 ms.

Glycopeptide fragmentation data were extracted from the raw file using Byos (Version 4.6; Protein Metrics Inc., Boston, MA, USA). The glycopeptide fragmentation data were evaluated manually for each glycopeptide; the peptide was scored as true positive when the correct b and y fragment ions were observed along with oxonium ions corresponding to the glycan identified. The MS data were searched using the Protein Metrics 38 insect N-glycan library. The relative amounts of each glycan at each site as well as the unoccupied proportion were determined by comparing the extracted chromatographic areas for different glycotypes with an identical peptide sequence. All charge states for a single glycopeptide were summed. The precursor mass tolerance was set at 4 ppm and 10 ppm for fragments. A 1% false discovery rate (FDR) was applied. The relative amounts of each glycan at each site as well as the unoccupied proportion were determined by comparing the extracted ion chromatographic areas for different glycopeptides with an identical peptide sequence. Glycans were categorized according to the composition detected. HexNAc(2)Hex(9−4) was classified as M9 to M4. Any of these compositions containing fucose were classified as fucosylated mannose (FM). HexNAc(3)Hex(5−6)X was classified as Hybrid with HexNAc(3)Fuc(1)X classified as Fhybrid. Complex-type glycans were classified according to the number of processed antenna and fucosylation. Complex-type glycans were categorized according to the number of N-acetylhexosamine monosaccharides detected that do not fit in the previously defined categories. If all of the compositions have a fucose, they were assigned into the (F) categories. As this fragmentation method does not provide linkage information, compositional isomers were grouped; so, for example, a triantennary glycan contains HexNAc5 but so does a biantennary glycan with a bisect. Any glycan containing at least one sialic acid was counted as sialylated.

### 2.11. Negative-Stain Electron Microscopy (NsEM)

Purified 329 SOSIP Env or SOSIP ferritin NPs or SOSIP-I53-50 NP proteins were diluted to 0.03–0.05 mg/mL in PBS before grid preparation. A 3 µL drop of diluted protein (~0.025 mg/mL) was applied to previously glow-discharged, carbon-coated grids for ~60 s, blotted and washed twice with water, stained with 0.75% uranyl formate, blotted, and air-dried. Between 30 and 50 images were collected on a Talos L120C microscope (Thermo Fisher) at 73,000 magnification and a 1.97 Å pixel size. Relion-3.1 [37] or Cryosparc v4.5.1 [38] was used for particle picking and 2D classification.

### 2.12. CryoEM Sample Preparation, Data Acquisition, and Data Analysis

A total of 3 µL of purified SOSIP I53-50 NP sample at a concentration of 0.5 mg/mL was applied onto a freshly glow-discharged (PLECO easiGLOW) 300 mesh, 1.2/1.3 C-Flat grid (Electron Microscopy Sciences, Hatfield, PA, USA). After 20 s of incubation, the grids were blotted for 3 s at 0 blot force and vitrified using a Vitrobot IV (Thermo Fisher Scientific) under 22 °C with 100% humidity. Single-particle Cryo-EM data were collected on a 200 kV Talos Arctica transmission electron microscope (ThermoFisher Scientific) equipped with a Gatan K3 direct electron detector behind a 20 eV slit energy filter. Multi-frame movies were collected at a pixel size of 1.1 Å per pixel with a total dose of 58.3 e/Å^2^ at a defocus range from −0.5 to −2.4 µm. In total, ~3142 cryoEM movies were motion-corrected by Patch motion correction implemented in Cryosparc v4.5.1 [38]. Motion-corrected micrographs were corrected for contrast transfer function using Cryosparc’s implementation of Patch CTF estimation. Micrographs with poor CTF fits were discarded using a CTF fit resolution cutoff of ~6.0 Å. Particles were picked using a Blob picker, extracted, and subjected to an iterative round of 2D classification. Particles belonging to the best 2D classes with secondary structure features were selected for two classes of ab initio reconstruction. Particles belonging to the best ab initio class were refined in non-uniform 3D refinement with per particle CTF and higher-order aberration correction turned on and applying Icosahedral (I1) symmetry to generate a cyoEM density map. The model for I53-50A and I53-50B nanoparticles (PDB:7SGE) was docked into the map using Chimera1.7.1 [39] fitting in the map function.

### 2.13. Rabbit Immunizations

Rabbit immunization was outsourced to a contract research organization (CRO) named Liveon Biolabs Private Limited, Bengaluru, Karnataka, India. The immunization studies described here were carried out on female naive New Zealand White rabbits of 2.0–2.5 kg and 4 months old. The use of animals for this study was approved by Liveon Biolabs Private Limited IAEC (IAEC-approved protocol No.: LBPL-IAEC-008-01/2021; study number: LBPL/NG-1736 (EF)). All immunization procedures complied with animal ethical regulations and protocols of the Liveon Biolabs Private Limited IAEC committee. Before housing the animals, the experimental room was decontaminated by fumigation and the microbial load was checked by using the settle plate method. The experimental room floor was mopped daily once. Rabbits were housed in an environment-controlled room at a temperature of 20 ± 3 °C and relative humidity of 30–70%. The photoperiod cycle was 12 h of light and 12 h of darkness. An adequate fresh air supply of 12–15 air changes/hour and a sound level of <80 dB were maintained in the experimental room. The relative humidity and maximum and minimum temperature in the experimental room were recorded once daily. The animals were observed for mortality and morbidity twice daily, i.e., once in the morning and once in the afternoon. The animals were observed at least once daily for clinical signs throughout the observation period. Individual animal body weight was measured before test item administration on Day 1 and weekly thereafter (±1 Day) for all the groups of animals during the treatment period.

For all immunogens (329 SOSIP, SOSIP–ferritin NPs, and SOSIP-I53-50 NPs), groups of four rabbits were given two intramuscular (IM) injections (2 × 250 µL) on both quadriceps at weeks 0, 4, and 20. The immunization mixture involved 20 µg of SOSIP trimers, or an equimolar amount presented as SOSIP–ferritin NPs (25 µg) or as SOSIP-I53-50 NPs (32 µg), formulated in AddaVax adjuvant (1:1 *v*/*v*). Dose calculations were based on the peptidic molecular weight of the proteins (thus disregarding glycans), which were calculated as described in [26,27]. Blood samples of the rabbits were collected at weeks 0, 4, 6, 16, 20, and 22. Local anesthetic cream (lidocaine 4%) was applied for 5 min on the ear before drawing blood from the marginal ear vein. At the end of the experimental period, all the animals were sacrificed by using a lethal dose of sodium thiopental injection. During the immunization of rabbits, bleed protocols, pain, and distress were minute and momentary and did not affect the health of the animals. Discomfort and injury to rabbits were limited, was although they are unavoidable when conducting scientifically valuable research.

### 2.14. HIV-1 Pseudovirus Generation

The HIV-1 pseudoviruses were produced in HEK293T cells as described previously [17] by co-transfecting the corresponding full HIV-1 gp160 envelope plasmid and a pSG3ΔEnv backbone plasmid. Briefly, 1 × 10^5^ cells in 2 mL complete DMEM (10% fetal bovine serum (FBS) and 1% penicillin and streptomycin antibiotics) were seeded per well of a 6-well cell culture plate (Costar) the day before transfection. For transfection, envelope (1.25 μg) and delta envelope (pSG3ΔEnv backbone) plasmids (2.50 μg) were mixed in a 1:2 ratio in Opti-MEM (Gibco), with a final volume of 200 μL per well, and incubated for 5 min at room temperature. Next, 3 μL of PEI-Max transfection reagent (Polysciences) (1 mg/mL) was added to this mixture before further incubation for 15 min at room temperature. This mixture was then added dropwise to the HEK 293T cells supplemented with fresh complete DMEM growth media and incubated at 37 °C for 48 h. Pseudoviruses were then harvested by filtering cell supernatants with 0.45 μm sterile filters (mdi), aliquoted, and stored at −80 °C until usage.

### 2.15. HIV-1 Neutralization Assays

The neutralizing activity of the rabbit immune sera was tested against autologous and heterologous pseudoviruses by performing neutralization assays as described previously [40,41]. Neutralization was measured as a reduction in luciferase gene expression after a single round of infection of TZM-bl cells (NIH AIDS Reagent Program) with HIV-1 Env pseudoviruses. The TCID_50_ of the HIV-1 pseudoviruses was calculated and 200 TCID_50_ of the virus was used in the neutralization assays through incubation with three-fold serially diluted rabbit sera starting at a 1:20 dilution. Next, freshly trypsinized TZM-bl cells in a growth medium (complete DMEM with 10% FBS and 1% penicillin and streptomycin antibiotics) containing 50 μg/mL DEAE Dextran at 10^5^ cells/well were added and the plates were incubated at 37 °C for 48 h. Virus controls (cells with HIV-1 virus only) and cell controls (cells without virus and antibody) were included. MuLV was used as a negative control. After the incubation of the plates for 48 h, luciferase activity was measured using the Bright-Glow Luciferase Assay System (Promega). ID_50_ for antibodies were calculated from a dose–response curve fit with a non-linear function using GraphPad Prism 9 software (San Diego, CA, USA). All neutralization assays were repeated at least 2 times, and the data shown are from representative experiments.

### 2.16. Statistical Analysis

GraphPad Prism version 9.0 was used for all statistical analyses.

## 3. Results

### 3.1. The Design and Biophysical Characterization of a Native-Like 329 SOSIP.v8.2 Env Trimer

We previously reported the isolation and characterization of multiple Env pseudoviruses from an Indian HIV-1 clade C seropositive pediatric elite neutralizer (AIIMS_329) whose plasma antibodies showed broad and potent HIV-1 neutralization in a longitudinal study [22,42]. One of these autologous Env pseudoviruses, 329.14.B1, showed exceptional susceptibility to neutralization by the majority of bnAbs in a panel covering multiple-epitope specificities and was resistant to non-nAbs and soluble CD4 (sCD4) (Appendix A). Hence, we selected this Env to stabilize it in soluble trimeric form. We engineered the 329 SOSIP.v8.2 Env trimer (Figure 1A) by introducing SOSIP mutations (501C-605C, 559P), including a multibasic furin cleavage site (hexa-arginine or R6) between gp120 and gp41, to generate cleaved 329 trimers. This protein also incorporated TD8 (47D, 49E, 65K, 165L, 429R, 432Q, 500R) [10] mutations that are well known to stabilize clade C HIV-1 soluble trimers and MD39 stabilizing mutations (106E, 271I, 288L, 304V, 319Y, 363Q, 519S, 568D, 570H, 585H) [31], which represents an advanced design of the HIV-1 Env trimer aimed at improving its stability and antigenic properties for use in vaccine development and immunological research. We also introduced a mutation to reduce V3 exposure (66R) [25]. We also introduced changes to optimize the epitope of HIV-1 trimer-specific bnAb PGT145 (166R, 168K, 170Q, 171K) as the 329 virus is resistant to PGT145 (Appendix A).

We expressed 329 SOSIP.v8.2 in HEK293F cells, followed by PGT145 antibody affinity chromatography purification (Figure 1B). The purification yield of this trimer was ~0.6 mg/L, which is comparable (0.4–0.6 mg/mL) to a previous clade C Env from a South African strain (CZA97.012, 0.4–0.6 mg/mL) [43]. Moreover, 329 SOSIP.v8.2 Env showed a single gp140 Env trimer band in BN-PAGE (Appendix A). In the binding ELISAs, 329 SOSIP.v8.2 Env interacted well with key HIV-1 bnAbs targeting major known epitopes including V1V2-apex (PGT145, CAP256.25, PG16), V3-glycan (2G12, AIIMS-P01, 10-1074, PGT121), CD4bs (VRC01, N6), fusion peptide (VRC34), and the gp120-gp41 interface (PGT151, 35O22) and showed negligible binding to all non-nAbs (19b, 14e, F240, b6, F105) tested, consistent with a native-like closed conformation (Figure 1C). The native-like conformation of stabilized 329 SOSIP.v8.2 Env well-ordered trimers was further confirmed by nsEM (Figure 1D). To determine the glycan composition of the 329 SOSIP.v8.2 Env trimer, we performed site-specific glycan analysis by mass spectrometry. Overall, the glycosylation profile of the 329 SOSIP.v8.2 clade C Env trimer presents a similar abundance of oligomannose-type glycan signatures at canonical “mannose sites”, including N160, N262, N332, and N448, as compared to previously characterized clade A and B Envs [26,27,28] (Figure 1E and Appendix A). To display the glycan holes or absence of potential N-linked glycosylation sites (PNGSs) (N289, N295, N339, and N386) in the 329 Env, a 3D model of the 329 SOSIP.v8.2 Env representing glycosylation was generated using AlphaFold 3 [44], GlycoShape, and Re-Glyco [45] (Figure 1F). Overall, the biophysical characterization suggests that the 329 SOSIP.v8.2 Env is successfully stabilized in a soluble native-like state and efficiently displays the epitopes for the key HIV-1 bnAbs tested in this study.

### 3.2. Displaying 329 SOSIP.v8.2 Env on Ferritin and Two-Component Nanoparticles

To increase the valency of 329 Env, we presented 329 SOSIP.v8.2 trimers on the surface of protein nanoparticles (NPs). First, we fused the C-terminus of 329 SOSIP.v8.2, after position 664, to the N-terminus of a previously described *H. pylori* ferritin [34] (GenBank accession no. NP_223316), starting at position Asp5 to Ser167, using a flexible Gly-Ser linker (GSG) (Figure 2A). The ferritin NPs can display eight native-like Env trimers [34]. Second, we previously described the computational design of two-component self-assembling I53-50 NPs [35], which are 120-subunit assemblies of icosahedral symmetry comprising 20 trimeric (I53-50A) and 12 pentameric (I53-50B) subunits. Therefore, each I53-50 NP can present up to 20 trimeric antigens fused to the I53-50A components. The possibility of purifying the I53-50A–antigen fusion proteins with trimer-selective purification methods before in vitro assembly with the I53-50B component ensures the presentation of native-like trimers exclusively. To further increase the valency of the 329 SOSIP.v8.2 Env, we genetically fused it to the I53-50A component (I53-50A.1NT1) via a Gly-Ser-rich linker (GSGGSGGSGGSGGS) (Figure 2B).

Next, the resulting SOSIP–ferritin and SOSIP-I53-50A fusion proteins were expressed in HEK293F cells, followed by purification using PGT145 bnAb-affinity chromatography (Figure 2C,D). The purified 329 SOSIP-I53-50A was mixed with the I53-50B component to fully assemble the 329 SOSIP-I53-50 NPs (Figure 2D). SEC purification of the fully assembled 329 SOSIP-I53-50 NPs revealed a peak at an elution volume of ~9.0 which is further shifted compared to the 329 SOSIP-I53-50A peak, indicating the formation of a high-molecular-weight complex (Figure 2D, right panel). The purified 329 SOSIP–ferritin and SOSIP-I53-50 NPs showed a single gp140 Env-displaying NP band in BN-PAGE (Appendix A). The purification yields for SOSIP–ferritin and the assembled SOSIP-I53-50 NPs were 0.6 mg/mL and ~1.0 mg/L, respectively. The fully assembled NPs when imaged by nsEM showed well-assembled NP structures with the 329 SOSIP.v8.2 Env trimer attached to the NP core (Figure 2E,F). These results confirm the multimeric presentation of the 329 SOSIP.v8.2 Env trimer on self-assembled ferritin and two-component I53-50 NPs. To confirm the proper assembly of I53-50 NPs and their structural integrity, we performed single-particle cryoEM analysis on 329 SOSIP-I53-50 NPs. The 3D model generated from the cryoEM data confirmed the construction of the I53-50 core (left) and multimeric presentation of 329 SOSIP trimers attached to each I53-50A moiety (right) (Figure 2G). Due to the high flexibility in the linker between the two domains, the 329 SOSIP trimers were poorly resolved and appeared as diffused densities surrounding the well-resolved I53-50NP core at a ~3.8 Å resolution.

### 3.3. 329 Env NPs Showed Improved Thermostability and Antigenicity

Next, we were interested in determining the effect of the multimeric display of 329 SOSIP.v8.2 on the stability and antigenic properties of self-assembled 329 SOSIP–ferritin and 329 SOSIP-I53-50 NPs. We first evaluated the thermostability of the proteins by using nano differential scanning fluorimetry (nanoDSF). Both SOSIP–ferritin and SOSIP-I53-50 NPs presented a higher T_m_ of 70.5 °C and 73.5 °C, respectively, as compared to the T_m_ of 68.1 °C observed for the 329 SOSIP.v8.2 Env trimer (Figure 2H). Next, we determined their antigenicity using Bio-Layer Interferometry (BLI). The V2-apex-targeting bnAbs PGT145 and CAP256.25 showed enhanced binding to both 329 SOSIP-I53-50 and SOSIP–ferritin NPs compared to the soluble 329 SOSIP.v8.2 Env trimer (Figure 3). The CD4bs-specific bnAb VRC01, N332-supersite-specific bnAb PGT121, and gp120-gp41 cleavage-specific bnAb PGT151, interacted efficiently with the 329 Envs. None of the non-nAbs (b6 and F105 against the CD4bs, and 19b targeting the V3) tested interacted with the soluble and multimerized 329 SOSIP.v8.2 trimers (Figure 3). Overall, these findings suggest that 329 SOSIP.v8.2 trimers maintain their native-like conformation in soluble form and when multimerized on ferritin NPs and I53-50 two-component NPs.

### 3.4. Multimerization on NPs Showed Higher Immunogenicity of 329 Env Trimers in Rabbits

Next, we compared the immunogenicity of the 329 SOSIP.v8.2 trimer with its 329 SOSIP–ferritin and 329 SOSIP-I53-50 NP variants in New Zealand White (NZW) rabbits. Four groups of four female rabbits were immunized at weeks 0, 4, and 20 with 20 µg of 329 SOSIP, or the equimolar amount presented on 329 SOSIP–ferritin and 329 SOSIP-I53-50 NPs, formulated in AddaVax™ adjuvant. Sera were collected from the rabbits at weeks 0, 4, 6, 16, 20, and 22 to assess the antibody responses (Figure 4A). First, we tested the binding titers of the collected immune sera from different timepoints to 329 SOSIP.v8.2 trimer using an ELISA (Figure 4B). An increase in binding titers occurred at week 6 and week 22, two weeks after the first and second booster immunizations, respectively (Figure 4B). We detected no significant differences in the binding titers induced by the different immunogens at any time point. Moreover, we evaluated the induced neutralizing antibody (nAb) titers against autologous AIIMS_329 (329.14.B1), closely related AIIMS_330 (330.16.E6) and heterologous clade C (25710 and MW965.26), Tier 1 clade B (SF162), and Tier 2 clade A (BG505.T332N) pseudoviruses (Figure 4C and Appendix A). Presentation on both ferritin and I53-50 NPs resulted in a higher trend of nAb titers against the autologous 329.14.B1 and the closely related 330.16.E6 pseudoviruses compared to the soluble format, although the differences were not statistically significant. The nAb titers induced by the SOSIP Env trimer against the other clade C Tier 1 viruses (MW965.26 and 25710) and a Tier 1 clade B virus (SF162) were comparable, which were not increased by NP immunizations. None of the 329 immunogens induced BG505-neutralizing responses, consistent with the low BG505 nAb activity observed in the plasma of the AIIMS_329 HIV-1-infected elite neutralizer [22]. Overall, these results demonstrate that both soluble and NP-displayed 329 SOSIP trimers induce autologous nAb responses, and that presenting them on NPs showed a higher trend of the induced nAb responses.

## 4. Discussion

Clade C HIV-1 Env trimers are generally unstable and challenging to stabilize for use in vaccines or capture reagents [10,11,18,20]. Most native-like soluble HIV-1 clade C trimers have been stabilized using cleavage-independent single-chain or native flexibly linked (NFL) platforms using clade A BG505 trimer-derived TD8 modifications [10,11]. However, these platforms show reduced binding to gp120-41-interface-targeting bnAbs such as PGT151, ACS202, and VRC34, limiting their use as antigenic baits for bnAbs discovery. HIV-1 Envs from elite neutralizers serve as valuable templates for designing vaccine candidates [3,4,20].

A recent analysis of clade C viruses from India and South Africa, countries with the highest HIV-1 disease burden due to clade C viruses, revealed that clade C sequences between these two countries differ significantly [46]. Such studies underscore that the distinct evolutionary trajectories of global intra-clade C Env sequences likely contribute to the region-specific sensitivity of circulating HIV-1 clade C to bNAbs. Hence, the design and characterization of geographically distinct clade C Env sequences with the ability to elicit protective elite HIV-1-neutralizing bnAb responses during natural infection are critical to further guide the rational design and development of globally effective vaccine candidates. Discovering novel HIV-1 bnAbs from clade C-infected donors including children and evaluating their quality, breadth, and epitope specificity could provide critical insights for effective HIV-1 vaccine design and vaccination strategies.

In recent years, multimeric presentation of viral class I fusion viral Env glycoproteins, such as Respiratory Syncytial Virus (RSV) F protein, influenza hemagglutinin (HA), Lassa virus (LASV) GPC protein, SARS-CoV-2 S protein, and HIV-1 Env, on ferritin and two-component protein nanoparticles (NPs) has enabled the efficient production of well-ordered NP-based vaccine candidates [26,27,28,34,47,48,49]. Here, we designed and stabilized native-like HIV-1 329 Env SOSIP trimers using soluble, ferritin, and two-component NP platforms. We introduced some of the newly reported HIV-1 Env stabilizing mutations identified to improve the stability and antigenic properties of the soluble trimers for use in vaccine development and immunological research [10,11,26,27,28,29,30,31]. These 329 SOSIP Env trimers were expressed and purified using PGT145 bnAb-affinity chromatography. The purified Env trimers showed reactivity to key HIV-1 bnAbs targeting multiple major epitopes and negligible binding to non-nAbs. The 329 SOSIP.v8.2 Env showed closed native-like conformation in nsEM analysis. Immunization studies in rabbits demonstrated a higher trend of the autologous nAb titers with both ferritin and two-component NPs than with the soluble 329 SOSIP.v8.2 Env trimer.

Although many HIV-1 Env immunogens have been stabilized and characterized using cleavage-independent (single-chain, NFL, and UFO) or cleavage-dependent (SOSIP) platforms, most have focused on the clade A BG505, a transmitted founder Env derived from a 6-week-old infant BG505, which is a key component of the ongoing HIV-1 clinical trials [9,10,11,12,24,25,29,30,50]. To the best of our knowledge, no such Envs were designed and characterized from an HIV-1 chronically infected pediatric elite neutralizer in a native-like soluble form.

In our immunization studies, the 329 Env trimer-induced nAbs were evaluated against a limited panel of pseudoviruses, as no significant enhancement in autologous or Tier 2 virus neutralization titers was observed. The observed relatively low autologous and heterologous neutralization ID_50_ titers may plausibly be due to the reduced antigenic exposure, the presence of complex glycan composition in 329 Env, or the use of the AddaVax adjuvant. Further, we detected lesser neutralizing activity of the immune sera of rabbits immunized with 329 SOSIP-I53-50 NPs for the autologous AIIMS_329 HIV-1 Env pseudovirus in comparison to a closely related AIIMS_330 Env pseudovirus, which could plausibly be due to the better exposure of neutralizing determinants on the AIIMS_330 virus or perhaps its higher susceptibility to HIV-1 nAbs [13,22]. Detailed studies to understand the effect of different adjuvants, the role of glycans, and the effect of various Env mutations are warranted to study their subsequent effects on immunogenicity.

Our work focuses on the stabilization of an Indian HIV-1 clade C trimer based on an Env sequence isolated from a pediatric elite neutralizer AIIMS_329. As stabilizing clade C trimers in general has been challenging, therefore, mutations MD39 and TD8 described for stabilizing the key vaccine candidate BG505 SOSIP from clade A and other HIV-1 clade C Envs, respectively, were applied to stabilize 329 SOSIP trimers using v8.2 mutations instead of directly incorporating just SOSIP mutations. All conclusions are based on our experiments performed in this study. The high-resolution structure of the 329 Env two-component NPs was not achieved due to the high flexibility in the linker between the two domains. The 329 Env trimers were tested against a limited panel of pseudoviruses as we did not observe enhanced autologous and Tier 2 virus neutralization titers in our rabbit immune sera experiments. Epitope mapping of the elicited rabbit sera antibodies was not performed. Further improvements in immunogenicity may be achieved through alternate adjuvants or design modifications.

## 5. Conclusions

Our findings show that presenting HIV-1 clade C trimers in a multimeric form could improve their overall stability, antigenicity, and immunogenicity in terms of eliciting higher autologous responses through the display of homogeneous arrays of native-like HIV-1 Env trimers. The 329 SOSIP.v8.2 Env designed and stabilized in this study could serve as a suitable antigenic bait for bnAb discovery and as an antigen for screening vaccine immune responses in future. The thermostable 329 SOSIP–ferritin and 329 SOSIP-I53-50 NPs can be components of multivalent immunogens aimed at eliciting multiclade specific or broad neutralizing antibody responses to HIV-1 in the future, especially in the population of low- or middle-income countries, and can facilitate vaccine distribution without any requirement of maintaining cold-chain storage conditions.

## Figures and Tables

**Figure 1 vaccines-13-00110-f001:**
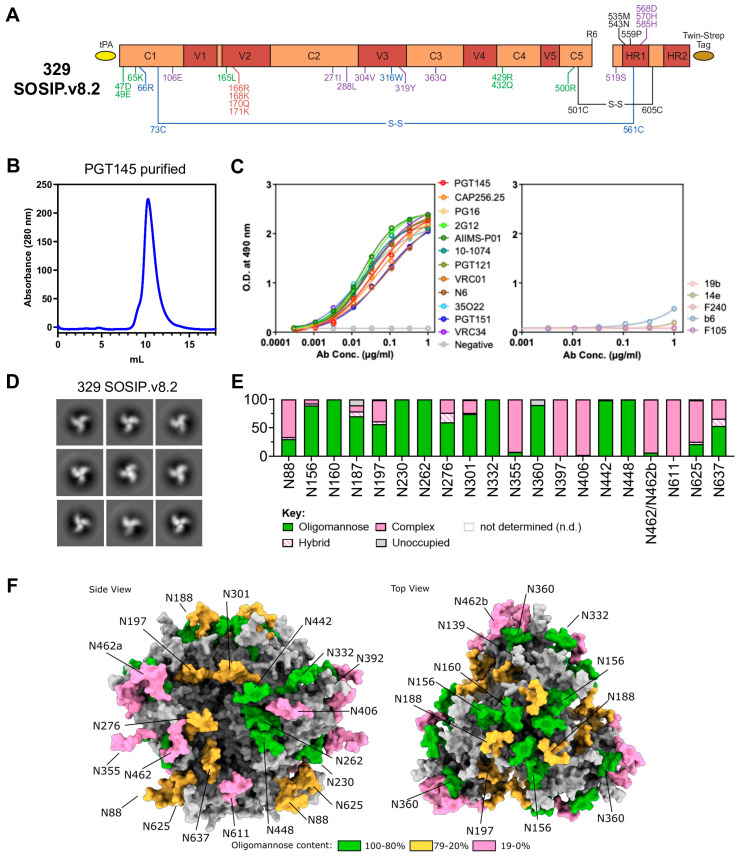
The design and biophysical characterization of the clade C 329 SOSIP.v8.2 Env trimer. (**A**) A linear representation of the 329 SOSIP.v8.2 construct, with SOSIP.664 mutations (501C-605C, 559P, R6) in black, further stabilizing SOSIP mutations (66R, 316W, 73C-561C) in blue, TD8 mutations (47D, 49E, 65K, 165L, 429R, 432Q, 500R) in green, MD39 mutations (304V, 319Y, 363Q, 519S, 568D, 570H, 585H) in purple, and PGT145 epitope modifications (166R, 168K, 170Q, 171K) in red. The following mutations (106E, 271I, 288L) that are part of MD39 stabilization are naturally present in 329 Env. (**B**) The SEC profile of PGT145-purified 329 SOSIP.v8.2 on a Superdex 200 Increase 10/300 GL column. (**C**) StrepTactinXT ELISA with PGT145-purified 329 SOSIP.v8.2 against a panel of bNAbs; V1V2-apex-targeting (PGT145, CAP256.25, PG16), V3-glycan-targeting (2G12, AIIMS-P01, 10-1074, PGT121), CD4bs-targeting (VRC01, N6), fusion peptide-targeting (VRC34), gp120-gp41-interface-targeting (PGT151, 35O22) (**left**), and non-NAbs (19b, 14e, F240, b6, F105) (**right**). (**D**) The 2D class averages generated from nsEM data of the PGT145-purified 329 SOSIP.v8.2 protein. (**E**) Site-specific glycan analysis of PGT145-purified 329 SOSIP.v8.2 protein. The values represented are specified in Appendix A. PNGSs are displayed as aligned with HxB2. The data could not be determined (n.d.) for sites N143, N241, and N397. The glycan modifications on the remaining sites were classified into three categories: high mannose (corresponding to any composition containing two HexNAc residues, or three HexNAc and at least five hexoses), complex, or unoccupied. The proportion of peptides and glycopeptides corresponding to each of these categories was colored green for high mannose, pink for complex, and gray for unoccupied. (**F**) The model of the glycan shield of 329 SOSIP.v8.2 generated using AlphaFold 3 and Re-Glyco. A representative Man_5_GlcNAc_2_ glycan is modeled at each site and colored according to the % oligomannose-type glycans displayed in panel E. Sites that could not be determined are colored gray.

**Figure 2 vaccines-13-00110-f002:**
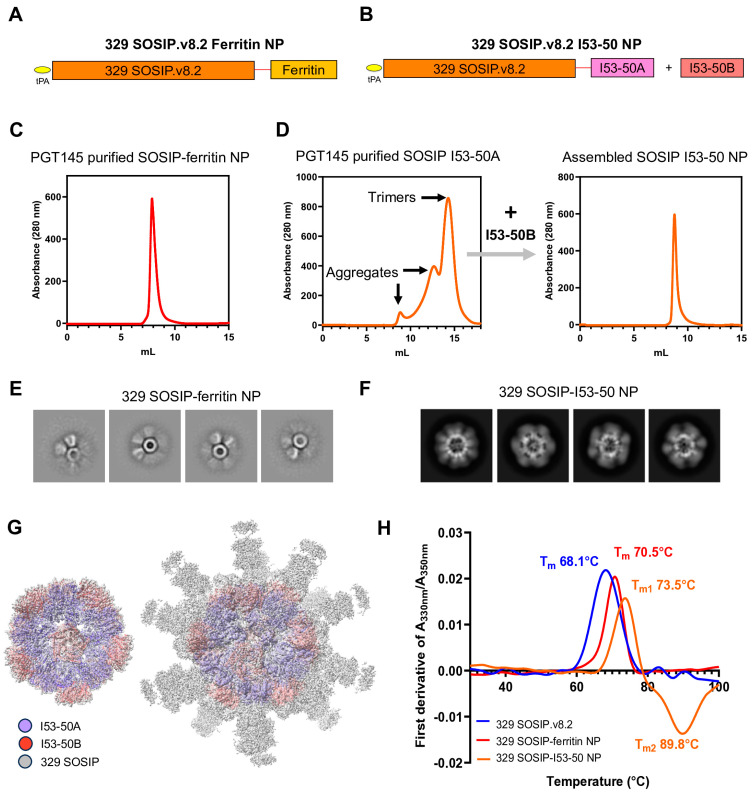
The design and biophysical characterization of 329 SOSIP–ferritin and SOSIP-I53-50 nanoparticles. (**A**,**B**) Linear representations of the 329 SOSIP–ferritin (**A**) and 329 SOSIP-I53-50A (**B**) fusion proteins. The red lines represent GSG and GSGGSGGSGGSGGS flexible linkers. (**C**) The SEC profile of PGT145-purified 329 SOSIP–ferritin NPs on a Superdex 200 Increase 10/300 GL column. (**D**) The SEC profile of PGT145-purified 329 SOSIP-I53-50A fusion protein (**left**) and 329 SOSIP_I53-50 assembled NPs (**right**) on a Superose 6 Increase 10/300 GL column. (**E**,**F**) nsEM-generated 2D class averages of 329 SOSIP–ferritin (**E**) and 329 SOSIP-I53-50 (**F**) NPs. (**G**) A 3.8 Å resolution cryo-EM map showing details of the I53-50 NP core (**left**), and the density of the 329 SOSIP trimers displayed on the I53-50 NP core (**right**) can be seen at a low contour level. (**H**) The denaturing profiles of 329 SOSIP, 329 SOSIP–ferritin NPs, and 329-SOSIP-I53-50 NPs, obtained by nanoDSF and used to determine the T_m_ values referred to in Section 3.

**Figure 3 vaccines-13-00110-f003:**
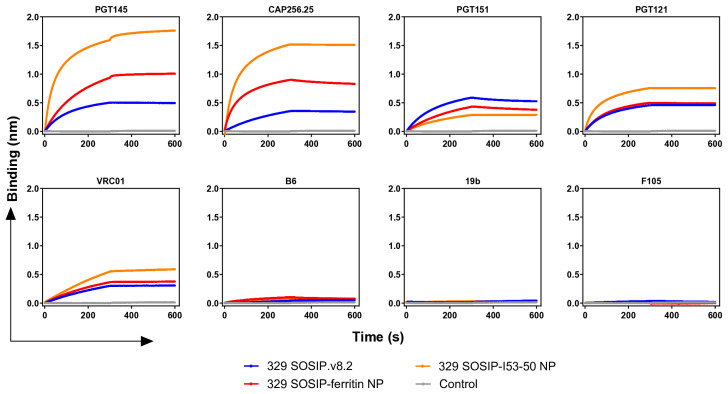
Antigenic analysis of 329 SOSIP, SOSIP–ferritin nanoparticles, and SOSIP-I53-50 nanoparticles. The ProtA BLI assay with 329 SOSIP, 329 SOSIP–ferritin NPs, and 329-SOSIP-I53-50 NPs and a panel of bnAbs (PGT145, CAP256.25, PGT151, PGT121, VRC01) and non-nAbs (B6, 19b, F105). The experiment was performed in duplicate, and the curves shown correspond to one of these repetitions. Association was measured for 300 s (0 to 300 s) and dissociation was measured for another 300 s (300 s to 600 s).

**Figure 4 vaccines-13-00110-f004:**
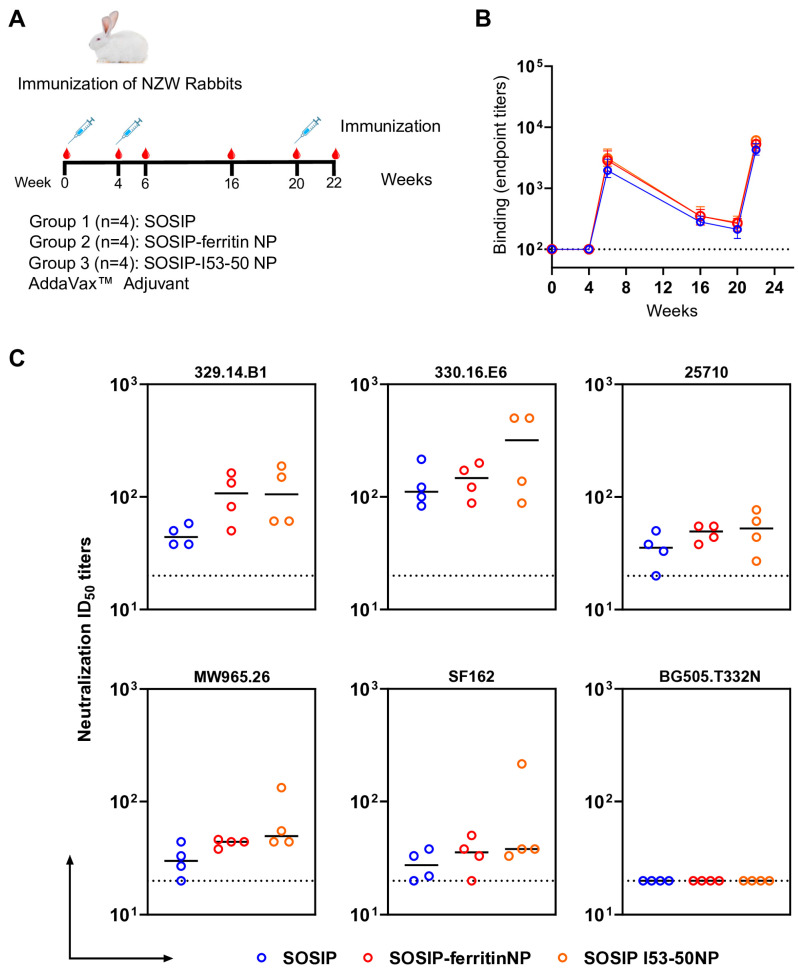
The immunogenicity of the 329 SOSIP.v8.2 Env trimer, SOSIP–ferritin NPs, and SOSIP-I53-50 NPs in rabbits. (**A**) The rabbit immunization schedule. Four groups of New Zealand White rabbits were immunized at weeks 0, 4, and 20 with 10 µg of SOSIP trimer (Group 1, n = 4) or equimolar amounts of SOSIP–ferritin NPs (25 µg) (Group 2, n = 4) or SOSIP-I53-50 NPs (32 µg) (Group 3, n = 4) or the placebo (PBS with Adjuvant) (Group 4, n = 2). Antibody responses were evaluated at weeks 0, 4, 6, 16, 20, and 22. (**B**) Endpoint antibody binding titers over time against the 329 SOSIP.v8.2 trimer as measured by StreptactinXT ELISA. The dots and error bars represent the median binding titers and standard deviations. No significant differences were found by a Kruskal–Wallis statistical test between groups at any time point tested. (**C**) Midpoint neutralization titers (ID_50_) for week-22 sera of the immunized rabbits against a panel of pseudoviruses. The horizontal lines represent the geometric means of the ID_50_ titers. No significant differences between groups were found by an unpaired two-tailed Mann–Whitney U-test. The assay cutoff is marked with a dotted line.

## Data Availability

The data supporting this manuscript may be available upon reasonable request to the corresponding author(s).

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
