# Peer review of "The Design and Immunogenicity of an HIV-1 Clade C Pediatric Envelope Glycoprotein Stabilized by Multiple Platforms"

_vaccines, 2025, doi:10.3390/vaccines13020110_

Round 1
Reviewer 1 Report
Comments and Suggestions for Authors
The manuscript is devoted to the hot spot of vaccine investigations, namely the preparation of HIV envelope protein immunogen capable of raising virus- neutralizing antibodies. Hence the manuscript fully matches the aims and scope of the Vaccines journal and is of interest to its readers. Its scientific soundness is indisputable and the research design is totally adequate and in general, but not in some details, well described. The latter has caused certain notes.
1. It is necessary to describe in more details two-component I53-50 particles. Despite the fact that these particles have been already described in another publication, their characterization in the presented manuscript will make the authors' idea of the employment of these particles for presenting and stabilizing HIV Env trimer constructs more clear. Differences in HIV Env presentation by the two NP platforms should be somewhat more clear emphasized, if the authors really observe them.
2. Sensograms of antibody binding with HIV Env trimeric constructs (Figure 3) present the association part of the curves. However, the specific binding is usually confirmed by the dissociation part of the curve. It is necessary to present the dissociation parts of the Ab binding curves, at least for PGT145 and CAP256.25 Abs. Otherwise there is no confirmation of the specific character of Ab binding.
3.Regarding the design of the 329.SOSIP.v8.2-ferritin construct, it is unclear, what part of the H. pylori ferritin N-terminal region has been used in it - "starting from Asp5", and terminating at which residue? (line 108).
4. Some experimental details sound confusing, as described. It seems unclear, what means the ration 1:30 (w/w), between what components (line 245);
"... with 1:3 serially diluted rabbit sera starting at 1:20 dilution." -sounds unclear at a glance, it is better to change this phrase to a more clear variant.
5. Certain experimental components are not described. The sources of Twin-Step-Tagged purified 329 SOSIP protein, (line 209), full and delta HIV envelope plasmids (lines 349, 353), neutralizing and non-neutralizing antibodies are not mentioned. "PGT145-functionalized CNBr-activated Sepharose..." (line 137-138) - PGT145 should be defined as a mAb, unless its use for HIV Enev protein purification is unclear.
6. Some abbreviations need a description: bnAbs (line 54), BN-PAGE (line 197), PNGS (line 416), sCD4 (line 387) SEC (line 427).
7. Table S3. It is unclear whether the title Clade corresponds to the column beneath or to the line to the right. What are the designations in the column beneath the title "Clade" mean if they don't represent the Clade?
Author Response
Comments and Suggestions for Authors
The manuscript is devoted to the hot spot of vaccine investigations, namely the preparation of HIV envelope protein immunogen capable of raising virus- neutralizing antibodies. Hence the manuscript fully matches the aims and scope of the Vaccines journal and is of interest to its readers. Its scientific soundness is indisputable and the research design is totally adequate and in general, but not in some details, well described. The latter has caused certain notes.
- It is necessary to describe in more details two-component I53-50 particles. Despite the fact that these particles have been already described in another publication, their characterization in the presented manuscript will make the authors' idea of the employment of these particles for presenting and stabilizing HIV Env trimer constructs more clear. Differences in HIV Env presentation by the two NP platforms should be somewhat more clear emphasized, if the authors really observe them.
Response: We appreciate the reviewer’s suggestion to provide a more detailed description of the two-component I53-50 nanoparticles. While these particles have been extensively characterized in previous publications, we agree that elaborating on their features in our manuscript will enhance the clarity of their application in presenting and stabilizing HIV Env trimer constructs.
In our study, we utilized two nanoparticle (NP) platforms for the display of HIV Env trimers: (1) Ferritin nanoparticles and (2) I53-50 two-component nanoparticles. We have described about the I53-50 NP in lines 836-844. We have observed notable differences in Env presentation between these two platforms, particularly in terms of trimer density and stability. The Ferritin NP platform involves the in-vivo assembly of HIV envelopes, resulting in the presentation of 8 copies of Env trimers on the particle's surface as observed by NS-EM (Figure 2E). In contrast, the I53-50 NP platform allows for greater control over the assembly process, which is conducted in vitro, and facilitates the presentation of 20 copies of Env spikes on the surface as observed by NS-EM (Figure 2F) and Cryo-EM (Figure 2G). Further, we observed a higher melting point Tm Figure 2H and enhanced binding affinity to bnAbs for both Ferritin and I53-50 NP Figure 3, where latter is superior.
- Sensograms of antibody binding with HIV Env trimeric constructs (Figure 3) present the association part of the curves. However, the specific binding is usually confirmed by the dissociation part of the curve. It is necessary to present the dissociation parts of the Ab binding curves, at least for PGT145 and CAP256.25 Abs. Otherwise there is no confirmation of the specific character of Ab binding.
Response: We appreciate the reviewer's comment regarding the importance of presenting both the association and dissociation phases in our Sensograms to confirm the specific binding of antibodies.
Figure 3 depicts the complete binding interaction, which includes both association and dissociation phases, measured using Biolayer Interferometry (BLI). Specifically, the association phase is captured from 0 to 300 seconds, while the dissociation phase spans from 300 to 600 seconds (Line 230). The quaternary bNAbs, PGT145, CAP256.25, and PGT151, demonstrated high affinity to the Env trimeric constructs, with notably stable dissociation curves. A stable dissociation is indicative of specific and robust binding, thus confirming the character of the antibody-Env interactions. We have updated this information in the Figure 3 legend.
- Regarding the design of the 329.SOSIP.v8.2-ferritin construct, it is unclear, what part of the H. pylori ferritin N-terminal region has been used in it - "starting from Asp5", and terminating at which residue? (line 108).
Response: The 329.SOSIP.v8.2-ferritin construct utilizes the H. pylori ferritin sequence from GenBank accession no. NP_223316), Asp5 to Ser167, fused to the truncated C-terminus of the SOSIP.664 trimer via a GSG linker, to create a nanoparticle that presents 8 copies of the 329 envelope trimer. This information is available in the Material and methods section under construct design heading (Line 105).
- Some experimental details sound confusing, as described. It seems unclear, what means the ration 1:30 (w/w), between what components (line 245);
"... with 1:3 serially diluted rabbit sera starting at 1:20 dilution." -sounds unclear at a glance, it is better to change this phrase to a more clear variant.
Response: Both the statements have been rephrased for better readability and understanding:
(1) The alkylated Env samples were buffer exchanged with 50 mM Tris/HCl, pH 8.0 using Vivaspin columns (10 kDa) and three aliquots were digested separately overnight, using trypsin, chymotrypsin (Mass Spectrometry Grade, Promega) or alpha lytic protease (Sigma Aldrich) at ratio of 1:30 (w/w) i.e. 1 µg enzyme: 30 µg Env protein (Line 247).
(2) 200 TCID50 of the virus was used in neutralization assays by incubating with three-fold serially diluted rabbit sera starting at 1:20 dilution (Line 371).
- Certain experimental components are not described. The sources of Twin-Step-Tagged purified 329 SOSIP protein, (line 209), full and delta HIV envelope plasmids (lines 349, 353), neutralizing and non-neutralizing antibodies are not mentioned. "PGT145-functionalized CNBr-activated Sepharose..." (line 137-138) - PGT145 should be defined as a mAb, unless its use for HIV Enev protein purification is unclear.
Response:
- We have replaced ‘the Twin-strep tagged purified’ with ‘PGT145 bnAb purified 329 SOSIP protein’ in Line 211.
- The delta HIV envelope (pSG3ΔEnv backbone) plasmid is described in line 372.
- We have provided the information of nAbs and non-nAbs in Figure 1 legend. The same has also been included in the revised text in line no 507 to 514
- Details of PGT145 bnAb (HIV-1 pre-fusion closed trimer specific mAb) is described in line 133
- HIV-1 pre-fusion closed trimer specific bnAb PGT145 was covalently coupled to the CNBr-activated Sepharose 4B beads (GE Healthcare/Cytiva) according to manufacturer’s protocol of antibody coupling. The PGT145-coupled CNBr Sepharose 4B beads column was used to purify HIV-1 Env trimers or Env trimer NPs (lines 132 to 136).
- Some abbreviations need a description: bnAbs (line 54), BN-PAGE (line 197), PNGS (line 416), sCD4 (line 387) SEC (line 427).
Response: Abbreviations have been expanded in the revised manuscript:
- bNAbs: Broadly Neutralizing Antibodies, Line 40
- BN-PAGE: Blue Native-Polyacrylamide Gel Electrophoresis, Line 200
- PNGS: Potential N-linked Glycosylation Sites, Line 419
- sCD4: Soluble CD4, Line 391
- SEC: Size Exclusion Chromatography, Line 160
- Table S3. It is unclear whether the title Clade corresponds to the column beneath or to the line to the right. What are the designations in the column beneath the title "Clade" mean if they don't represent the Clade?
Response: Table S3 has been modified as suggested in the revised manuscript supplementary file.
Reviewer 2 Report
Comments and Suggestions for Authors
Although the work have a good scientific soundness, it is quite difficult to follow in the description of the methods. And, consequently, in the conclusions. In details, the author could improve the work by adding a general scheme (graphical abstract like) to make evident the main objective of the research. Some detail on the background should be added to make the text understandable also to those reader who are not specialized in this specific research field.
Comments on the Quality of English Language
Also, some sentences are too long and hard to follow. The reviewer suggests a general revision of the clarity of the text.
Author Response
Although the work have a good scientific soundness, it is quite difficult to follow in the description of the methods. And, consequently, in the conclusions. In details, the author could improve the work by adding a general scheme (graphical abstract like) to make evident the main objective of the research. Some detail on the background should be added to make the text understandable also to those reader who are not specialized in this specific research field.
Response: Thank you for appreciating our work and providing your valuable comments. We have extensively simplified and provided details in the methods and text of the whole manuscript including conclusions. We have included a graphical abstract of the study and modified the ‘background’ in the abstract for better understanding.
Comments on the Quality of English Language
Also, some sentences are too long and hard to follow. The reviewer suggests a general revision of the clarity of the text.
Response: We have done a thorough revision of the text to improve the quality of the english language, ensuring that sentences are concise and easier to follow. Complex sentences have been simplified or broken down into shorter sentences to enhance clarity and readability. We believe these revisions have significantly improved the overall flow of the manuscript.
Reviewer 3 Report
Comments and Suggestions for Authors
Kumar et al generated a novel HIV-1 vaccine platform using a soluble clade C Env trimer isolated from a pediatric elite controller. Their multimerized platform exhibited high binding to broadly neutralizing HIV-1 antibodies with negligible binding to non-neutralizing antibodies. Further testing demonstrated that their multimerized platform had higher immunogenicity in rabbits compared to the “native” Env trimer. Taken together, this is a well-written and interesting study that could have important implications for future HIV-1 vaccine designs. I have only minor comments/concerns detailed below.
Major:
- Several of the figures and figure keys are blurry. For example, The diagram in Figure 1A, the key in Figure 1E, structural models and residue labels in Figure 1F, structural model in Figure 2G, etc.
Minor
- SOSIP is not defined
- I think all Figure 3 references from lines 516-539 should be changed to Figure 4 references
Author Response
Kumar et al generated a novel HIV-1 vaccine platform using a soluble clade C Env trimer isolated from a pediatric elite controller. Their multimerized platform exhibited high binding to broadly neutralizing HIV-1 antibodies with negligible binding to non-neutralizing antibodies. Further testing demonstrated that their multimerized platform had higher immunogenicity in rabbits compared to the “native” Env trimer. Taken together, this is a well-written and interesting study that could have important implications for future HIV-1 vaccine designs. I have only minor comments/concerns detailed below.
Major:
- Several of the figures and figure keys are blurry. For example, The diagram in Figure 1A, the key in Figure 1E, structural models and residue labels in Figure 1F, structural model in Figure 2G, etc.
Response: High resolution figures have been included in the PDF format, along with the revised manuscript file.
Minor
- SOSIP is not defined
Response: Description of the SOSIP mutations is included in line 393 “(SOS refers to a disulfide bond between the gp120 (501C) and gp41 (605C) region, and IP refers to the I559P mutation)”.
- I think all Figure 3 references from lines 516-539 should be changed to Figure 4 references
Response: We have updated all references from Figure 3 to figure 4 in lines 870-900.
Round 2
Reviewer 1 Report
Comments and Suggestions for Authors
No comments.